# A Novel NOX Inhibitor Treatment Attenuates Parkinson’s Disease-Related Pathology in Mouse Models

**DOI:** 10.3390/ijms23084262

**Published:** 2022-04-12

**Authors:** Anurupa A. Ghosh, Dinesh Kumar Verma, Gabriela Cabrera, Kwadwo Ofori, Karina Hernandez-Quijada, Jae-Kwan Kim, Joo Hee Chung, Michael Moore, Sung Hwan Moon, Jong Bok Seo, Yong-Hwan Kim

**Affiliations:** 1Department of Biological Sciences/Neuroscience Program, Delaware State University, Dover, DE 19901, USA; dverma@desu.edu (D.K.V.); gncabrera19@students.desu.edu (G.C.); kofori19@students.desu.edu (K.O.); khernandez18@students.desu.edu (K.H.-Q.); 2Seoul Center, Korea Basic Science Institute, Seongbuk-gu, Seoul 02841, Korea; jkvision1101@kbsi.re.kr (J.-K.K.); jjh@kbsi.re.kr (J.H.C.); sjb@kbsi.re.kr (J.B.S.); 3Imaging Core, Delaware State University, Dover, DE 19901, USA; mmoore@desu.edu; 4AptaBio Therapeutics Inc., 504 Tower, Heungdeok IT Valley, Heungdeok 1-ro 13, Gyeonggi-do, Yongin-si 16954, Korea; shmoon1978@aptabio.com

**Keywords:** α-synuclein preformed fibrils, ROS inhibition, Thioflavin T, oxidative stress, protein aggregation

## Abstract

Parkinson’s disease (PD) is a progressive neurodegenerative motor disorder without an available therapeutic to halt the formation of Lewy bodies for preventing dopaminergic neuronal loss in the nigrostriatal pathway. Since oxidative-stress-mediated damage has been commonly reported as one of the main pathological mechanisms in PD, we assessed the efficacy of a novel NOX inhibitor from AptaBio Therapeutics (C-6) in dopaminergic cells and PD mouse models. The compound reduced the cytotoxicity and enhanced the cell viability at various concentrations against MPP+ and α-synuclein preformed fibrils (PFFs). Further, the levels of ROS and protein aggregation were significantly reduced at the optimal concentration (1 µM). Using two different mouse models, we gavaged C-6 at two different doses to the PD sign-displaying transgenic mice for 2 weeks and stereotaxically PFF-injected mice for 5 weeks. Our results demonstrated that both C-6-treated mouse models showed alleviated motor deficits in pole test, hindlimb clasping, crossbeam, rotarod, grooming, and nesting analyses. We also confirmed that the compound treatment reduced the levels of protein aggregation, along with phosphorylated-α-synuclein, in the striatum and ventral midbrain and further dopaminergic neuronal loss. Taken together, our results strongly suggest that NOX inhibition can be a potential therapeutic target for PD.

## 1. Introduction

Parkinson’s disease (PD) is an irreversible neurodegenerative condition, most prevalent in the population over the age of 65 years. It is associated with dopaminergic neuronal death in the Substantia Nigra pars compacta (SNpc), which results in complex motor and non-motor deficits [1,2,3]. Although extensive research is warranted to understand PD pathology, there still exists a large consensus about the hallmarks of PD pathology at histological levels, such as (1) the increased levels of phosphorylated Ser-129-α-synuclein, (2) the presence of Lewy bodies, and (3) the irremediable loss of dopaminergic neurons [1,2,3,4]. This trifecta occurrence ultimately causes reduced or depleted dopamine levels and motor deficits in PD patients. A myriad of reasons has been pointed out to be causative of PD pathology. For example, some of them are exposure to neurotoxins, genetic defects, aging, cellular senescence, etc.; however, ultimately oxidative stress often mediates the upstream causes for inducing neuronal death in the midbrain [5,6,7]. The existing treatments for PD only suffice to normalize dopamine levels but do not target to halt neuronal loss. Therefore, an imperative need exists in developing therapeutic molecules to retard PD progression and alleviate its symptoms.

Reactive oxygen species (ROS) is a broad definition used for free radicals and reactive molecules generated from different cellular and molecular processes. Even though ROS is central to cell signaling via the redox pathway in various post-translational modifications, a plethora of evidence has emerged on the impact of elevated ROS generation, known as “oxidative stress” and cellular damage on mediating neurodegenerative progression [8,9,10,11,12]. The chronic state of increased cellular oxidative stress causes detrimental pathological conditions, leading to tissue damage and functional deficits [7,12]. A pronounced buildup of protein aggregates in certain sub-population of neurons has been associated with mitochondrial dysfunction due to toxic levels of oxidative stress, ultimately leading to irreversible cell death in the central nervous system (CNS) [10,12]. Since oxidative stress is a critical mechanism to trigger a neuronal loss in PD pathology, there is an evident need to target the pathway that successfully mitigates the levels of cellular oxidative stress to prevent neurodegeneration. The ROS production in cells can be either endogenous or exogenous. Among the exogenous sources, the main ROS contributors are nicotinamide adenine dinucleotide phosphate (NADPH) oxidase (NOX), lipoxygenase, cyclooxygenase (COX), xanthine oxidase, cytochrome P450, and nitric oxide synthetase [10]. When NOX activation is in conjunction with augmented oxidative stress, the NOX-induced ROS generation can cause substantial cell/tissue damage, if it is not resolved within a reasonable period [9,11]. In the midbrain, ROS is mainly generated during dopamine metabolism or when the level of glutathione (GSH) in the SNpc is low and/or the levels of iron and calcium are elevated [8,13,14]. It is thus noteworthy to target ROS inhibition for validating that a potent and specific NOX inhibitor can be an effective therapeutic in halting PD pathology by rescuing dopaminergic neuronal loss.

As reported, ROS generation is substantially high when the levels of nicotinamide adenine dinucleotide phosphate (NADPH) oxidases (NOX) are elevated [5,14,15]. The NOX family comprises transmembrane proteins usually responsible for the reduction of oxygen to superoxide from electron-donating cytosolic NADPH through flavin adenine dinucleotide (FAD) [16,17]. There are seven different NOX isoforms, out of which only NOX-1, -2, and -4 have been shown to be expressed in neurons, and the expression of NOX-2 has been reported high in non-neuronal cells during brain injuries [18,19,20,21]. Here, we assessed whether the inhibition of NOX-1,-2 and -4 (Table 1) can be beneficial in reducing cellular oxidative stress and preventing protein aggregation, and further dopaminergic neuronal loss in PD pathology for opening up new avenues of targeted therapies [14,15]. The first-in-class pan-NOX inhibitor, Isuzinaxib (APX-115) was designed and developed by AptaBio Therapeutics Inc. (Yongin, S. Korea), initially as a treatment for renal failure and tissue damage [22].

The purpose of this study is to test the feasibility of the C-6 in different PD in vitro and mouse models for developing a potential new therapeutic for PD. In this study, the efficacy of the C-6 was tested in MPP+ or preformed fibrils of α-synuclein (PFFs) exposed in vitro models, α-synuclein wild-type overexpressing transgenic [23,24] and PFF-injected mouse models [25,26]. Our results demonstrated that the NOX inhibitor alleviated the PD-like signs in behavior due to the increased dopamine release in the transgenic mouse model [26], in addition to reducing the PFF-induced protein aggregation and pathological conditions in mice. Therefore, our results strongly suggest that NOX inhibition with the C-6 can be an effective therapeutic application to prevent or possibly reverse PD pathology [14,15,27].

## 2. Results

### 2.1. The C-6 Enhances Cell Viability and Reduces Cytotoxicity from MPP+ or PFF

The MPP+ treatment at 640 µM for 24 h causes a sharp decrease in cell viability and increases the level of cytotoxicity in N27 dopaminergic cells [28]. The reduced cell viability was rescued by the C-6 treatment in a dose-dependent manner, up to 1 µM (Figure 1A), although 10 nM or lower were not significant, and the same results were obtained in the cytotoxicity analysis (Figure 1B). The optimal concentration of the compound was 1 µM in both MTT and LDH assays in the tested range of 1 nM to 10 µM, in which the cell viability increased by almost 50% and the cytotoxicity decreased by 50% in comparison with the MPP+ only group (Figure 1A,B). The average of the “No MPP+” group was taken as 100%, and the other groups were compared with it for relativity. The NOX inhibitor enabled to reverse MPP+-induced toxicity in N27 dopaminergic cells.

Next, we applied the preformed fibrils (PFFs) of α-synuclein to assess the protective effects of the C-6 in N27 cells. As previously reported, the optimal concentration of PFF to induce cytotoxicity in vitro is 1 µg/mL [28,29,30]. After 48 h incubation with PFF, the treatment groups received the log scale of the C-6 ranging from 1 nM to 10 µM. The remaining two groups of “no PFF” and “PFF only” are considered positive and negative controls, respectively. The C-6 significantly mitigated the PFF-induced cytotoxicity in the range of concentrations (100 nM to 1 µM) consistently in both MTT and LDH assays; however, the optimal dose in both assays was 1 µM (Figure 1C,D). Thus, the optimal concentration of 1 µM was set for the following assays such as ROS and Thioflavin-T staining.

### 2.2. The C-6 Treatment Reduces the Levels of ROS and Thioflavin-T Induced by PFF (or MPP+)

To assess the mechanisms of cytoprotection, we assessed whether the optimal dose of the compound reduces the ROS level induced by MPP+ (Figure 2A) or PFF-triggered protein aggregation in N27 cells. The 24 h exposure of MPP+ (640 µM) caused about 60% increase in ROS generation; however, about 40% of ROS was recovered within 24 h of the C-6 treatment (Figure 2B). Next, we measured the levels of ROS generation and Thioflavin-T-labeled protein aggregation with the treatment, after N27 cells were exposed to 1 µg/mL PFF for 24 h. All groups received WGA (5 µg/mL) and N-acetyl glucosamine (0.1 M), and cells were treated with the C-6 and incubated for 48 h. Our image analysis showed that ROS generation was also increased proportionally to the protein aggregation as stained in Thioflavin-T (Figure 3A). The protein aggregates in the PFF-treated group were 60% higher than those in both no-PFF and the C-6-treated groups (Figure 3B). Further, there was an over 2.5-fold increase in ROS generation with PFF exposure, whereas the level of ROS was substantially reduced with the C-6 treatment, which was nearly as low as the level of no PFF (Figure 3C). The C-6 treatment at 1 µM significantly reduced protein aggregation and ROS generation derived from PFF (Figure 3B,C).

### 2.3. The Novel NOX Inhibitor Alleviates Behavioral Deficits in L-61 Transgenic Mice

As the experimental scheme in Figure 4A illustrates, male transgenic mice (Line 61) were aged three months, and their motor skills were assessed in several behavioral tests, such as hindlimb clasping, latency on rotor road, nesting capabilities, crossbeam, and pole-test analyses. Before oral treatments, the transgenic mice showed typical motor deficits when compared with the wild-type littermates (Figure 4B–F). Based on our data analyses in a blinded manner using paired Student’s *t*-tests, we consistently found that two weeks of daily oral treatment at the high dose of 25 mg/kg enhanced motor functions, compared with the same mice before gavaging (*n* = 7–8/group) (Figure 4B–F). As evident in Figure 4B, the transgenic mice showed low latency on the rotarod, compared with the wild-type littermate, whereas the compound treatment enhanced latency on the rotarod in the transgenic mice. After the completion of two weeks of gavaging, there was a significant improvement in their latency time on the rotarod for both low and high doses. There was also a decrease in the hindlimb clasping time with the high-dose treatment (Figure 4C). Similar results were obtained in the nesting test, according to which nest-building capabilities were improved in post-gavaging (Figure 4D). The transgenic mice also showed fewer foot slips in the crossbeam assessment and a decrease in climb-down time in the pole test with the higher dose of treatment: 25 mg/kg (Figure 4E,F). Before treatment was labeled as pre-gavaging and after treatment as post-gavaging, and each value pertaining to scores obtained from an individual mouse for every group was compared in pair (pre- vs. post-gavage).

### 2.4. The Compound-6 Treatment Enhances Dopamine Release into the striatum in L-61 Transgenic Mice

As previously reported [26,31], the L-61 transgenic mice do not show a significant loss in the nigrostriatal neurons in the striatum and the midbrain. In our own analyses, the TH+ levels in both the striatum and the SNc were not significantly different between the hemizygous transgenic and littermate WT mice (Appendix A). However, the treatment of 25 mg/kg enhanced the level of dopamine released into the striatum, which was measured in liquid chromatography (LC) coupled with mass spectrometry using striatal extracts (Figure 5). The substantial increase in dopamine release with the treatment supports the results that behavioral deficits from the transgenic mice were alleviated in post-gavaging, demonstrated in Figure 4B–F.

### 2.5. The Oral Treatment Reduces the Levels of Protein Aggregates and Phosphorylated Ser-129-α-Synuclein in the L-61 Transgenic Mice

The level of phosphorylated Ser-129-α-synuclein (p-Ser-α-syn) was significantly higher in the transgenic mice than in their littermate WT mice. In Thioflavin-T staining, there was also a significant increase in the level of protein aggregation from the transgenic vehicle group than WT vehicle mice (*n* = 7–8/group, Figure 6A). In the immunohistochemistry (IHC) data analyses, the oral treatment of the C-6 at 25 mg/kg significantly decreased the levels of protein aggregates (Figure 6B) and phosphorylated Ser-129-α-synuclein (Figure 6C) in the striatum of transgenic mice. To further quantify the expression levels of phosphorylated Ser-129-α-synuclein in both the striatum and the ventral midbrain (vMB), we isolated the tissues from the left hemisphere of each brain in all four groups. In Western blot analyses, there was a significant increase in phosphorylated Ser-129-α-synuclein in transgenic mice with vehicle only, compared with WT vehicle. (Figure 7A,B). With the treatment at the high dose, the phosphorylated form of Ser-129-α-synuclein was decreased to nearly the level of the healthy/wild-type group with the vehicle in STr and vMB (*n* = 7–8/group, Figure 7C,D).

### 2.6. The Compound Treatment Improves from the Behavioral Deficits in PFF-Injected Mice

As shown in Figure 8A, an oral treatment for 5–6 weeks was initiated 4 ½ months after PFF injection, followed by behavioral tests (*n* = 7–8/group). We observed an improved latency on the rotarod in PFF-injected mice treated with both low- and high-dose treatments (Figure 8B). The other behavioral analysis showed that the PFF-injected mice were also improved in nesting capabilities with the compound treatment at the high dose (Figure 8C). In addition, the high-dose treatment showed a significant improvement in grooming, whereas the PFF-only controls showed poor grooming capabilities on a scale (out of 10) by a blinded rater (Figure 8D). The climb-down time in the pole test was decreased, along with a significant reduction in the hindlimb clasping, for both groups receiving the low or high dose of the compound (Figure 8E,F). These results suggest that the compound enabled the prevention of a decline in motor deficits or even reversal of dysfunctions for mice injected with PFF when compared with the vehicle-treated PFF-injected group (*n* = 7–8/group).

### 2.7. The Compound Treatment Prevents or Even Reverses PFF-Mediated Dopaminergic Neuronal Loss in the Striatum and the SNc

Five months after the stereotaxic PFF injection in the dorsal striatum, PFF consistently induced dopaminergic neuronal death in the striatum and the SNc, based on the IHC analyses, performed in a blinded manner (Figure 9A). Five-six weeks of daily oral treatment (25 mg/kg) starting 4 ½ months after PFF injection recovered and/or prevented further damage from PFF in the striatum and SNc, compared with PFF-only mice (*n* = 7–8/group). As seen in Figure 9B,C, only 25 mg/kg treatment showed a significant increase in TH+ intensity in the striatum and rescued or prevented the loss of dopaminergic neurons in the SNc region from PFF-induced toxicity. Although the low dose displayed a trend clearly and some effects in behavioral tests, only the 25 mg/kg dose was consistently significant in our behavioral tests and IHC analysis.

### 2.8. The Compound Treatment Reduces the Levels of Phosphorylated Ser-129-α-Synuclein and Protein Aggregates in PFF-Injected Brains

The results of Western blots revealed that the levels of Ser-129-α-synuclein in both the striatum and the ventral midbrain (vMB) were substantially increased in PFF-injected mice, compared with vehicle controls (*n* = 7–8/group, Figure 10A,B). In the following data analyses, we found that both 5 mg/kg and 25 mg/kg treatments reduced the levels of Ser-129-α-synuclein accumulation in both the striatum and SNc regions (Figure 10C,D). In a further analysis using IHC, the compound treatment at both doses decreased the levels of protein aggregation in the striatum based on Thioflavin-T staining and reduced the levels of Ser-129-α-synuclein in the region (Figure 11A). Our results demonstrated that both high and low doses of the compound were significantly efficacious in decreasing the levels of the pathological indicators in PFF-injected brains, since the toxic effects were correlated with the high levels of protein aggregates in thioflavin-T stain and Ser-129-α-synuclein, which were significantly lower in both treated groups (5 and 25 mg/kg) (Figure 11B,C).

### 2.9. The compound-6 Inhibits the Expressions of NOX-1 and NOX-2 in the Striatum of PFF-Injected Mice

As shown in Table 1, the C-6 is considered a NOX-1, -2, and -4 inhibitor. Here, we measured the levels of NOX-1, -2, and -4 from the striatal or ventral midbrain tissues in Western blots (Figure 12A,B). We found that PFF injection enhanced the levels of NOX-1 and -2 in the striatum and vMB, although the level of NOX-4 was consistently low in brains (data not shown). As expected, NOX-2 was detected around 55 kD, but two bands of NOX-1 were detected at 59 kD and 63 kD (probably glycosylated form) [32,33]. Our data analysis indicated that the levels of NOX-1 and -2 were significantly reduced by the treatment in both regions with both low- and high-dose treatment (Figure 12C,D).

## 3. Discussion

In this study, we demonstrated that the novel compound-6 was effective in halting or reversing N27 rat dopaminergic cell death by decreasing cytotoxicity and increasing cell viability against toxic agents, such as MPP+ and PFF. As previously reported, PFF-induced toxicities also enhance protein aggregation and increase the levels of ROS generation in N27 rat dopaminergic cells [28]. The investigation of ROS generation and co-labeling with Thioflavin-T further supported that the compound sufficiently reduced the levels of oxidative stress and protein aggregation in vitro. Here, our experimental approaches were focused on testing the recovery effects of the C-6 at various concentrations, ranging from 1 nM to 10 µM, in N27 rat dopaminergic cells, since the cells were first exposed to toxic agents, either PFF or MPP+, and followed by the compound treatment (Figure 1, Figure 2 and Figure 3). The treatment of MPP+ or PFF in N27 cells likely causes significant mitochondrial damage, resulting in poor cell viability [28], but we found an increase in cell viability with the C-6 treatment (Figure 1). Further, the compound consistently reduced the levels of ROS against MPP+ and PFF toxicities and also reduced the level of protein aggregation induced by the PFF treatment in N27 cells. The NOX inhibitor thus enabled us to show protective and some level of recovery effects in dopaminergic cells against ROS generating neurotoxic agents. Thus, the inhibition of NOX was successful in preserving mitochondrial function and preventing cell loss by reducing ROS generation. Based on our initial assessment using the PAMPA assay, the C-6 passed the artificial BBB effectively (Appendix A) and suppressed the activities of NOX-1, -2, and -4 effectively in IC_50_ analysis (Table 1).

Therefore, we assessed the effects of the C-6 treatment using a transgenic mouse model known as the L-61 mouse [31]. The transgenic mice overexpressing full-length human wild-type α-synuclein under the Thy-1 promoter provided us with a useful tool to test if the compound reduces their PD-related motor deficits [24,34]. The male transgenic mice showed PD-related behavioral signs along with region-specific synucleinopathy as early as 3–4 months. However, they did not show significant neuronal losses in the SNc region or a decrease in TH intensity in the striatum, which is verified in Appendix A [24,27,29,34]. The motor deficits are probably derived from the accumulated levels of α-synuclein, especially in the striatal region, in which these mice experience disruptions in dopamine release and reuptake [24,29,30,35]. Although there was no obvious neuronal loss in L-61 mouse brains, the dysregulation of dopamine released into the striatum caused severe motor deficits, due to the increase in protein aggregates [24,30,34,35]. In the following behavioral analyses, motor dysfunctions including the increased hindlimb clasping due to low levels of dopamine and/or dopaminergic neuronal loss were rescued by the C-6 treatment in vivo [23,24,36,37]. Other behavioral analyses of pole test, nesting, crossbeam, and rotarod analyses further supported the improved motor skills with the C-6 treatment at 25 mg/kg (Figure 4). Using this mouse model, we were able to provide supporting results of the C-6 efficacy as a recovery agent, as it alleviated motor dysfunction by enhancing dopamine release in the striatum, which was mechanistically supported by reduced protein aggregates and phosphorylated α-synuclein accumulation in the striatum, as seen in immunohistochemistry and Western Blots.

Since the L-61 mice do not show typical neurodegeneration in the nigrostriatal neurons [23,24], we decided to include another PD mouse model, PFF-injected mice for testing the potential efficacy of the compound [30,38,39]. Several previous studies have shown the toxicity caused by PFF can lead to neuronal loss and severe motor deficits in C57Bl/6 WT mice [29,40,41]. The PFF injection in the dorsal striatum induces the aggregation and accumulation of various proteins including endogenous α-synuclein in mouse brains. These mice also show protein aggregates containing phosphorylated-Ser129-α-syn inclusions, which are further associated with the loss of dopaminergic neurons in the midbrain [29,30,36,38,39]. Five weeks of the C-6 oral treatment at both low and high doses significantly mitigated the protein aggregation and further provided neuroprotection against toxic α-synuclein-mediated protein aggregation [40]. The oral gavaging of the C-6 showed that the NOX inhibition was successful in preventing dopaminergic neuronal loss in the PFF-injected mice. Our results demonstrate that NOX inhibition prevented the additional neuronal loss in the SNc region, as previously seen in 5–6 months post-PFF injection [28,40]. The inhibition of NOX can be a valid therapeutic target, since the compound effectively intervened, and even reversed the PFF-mediated protein aggregation and further cell death in nigrostriatal neurons.

Most PD pathology-mimicking mouse models have been attributed to proteinopathies with the accumulation of extensive endogenous levels of α-synuclein, resulting in neuronal loss. Both mouse models in our study include the accumulation of α-synuclein in causing protein aggregation and cellular dysfunction to mimic PD-related pathology [26,29,39,41]. In many cases, however, the protein aggregation *per se* is not sufficient enough to trigger neurodegeneration in mouse models, even though there is compelling evidence between oxidative stress and protein aggregation-mediated synucleinopathy involved in neurodegeneration [42]. Therefore, our approach is designed to target the downstream in suppressing the level of oxidative stress for preventing or possibly reversing neurodegeneration in some extent [40,42]. The C-6 successfully reduced the levels of insoluble protein aggregates, as seen in the IHC analysis from both transgenic and PFF-injected mouse models, but also decreased the levels of phosphorylated Ser-129-α-synuclein (~19 kDa), as seen in both Western blot and IHC analyses (Figure 10 and Figure 11). The excessive level of oxidative stress, primarily due to protein aggregation, gave rise to dopamine loss and related severe motor deficits, as seen in the cumulative analysis from both mouse models [27,29,30,38,41]. Behavioral analyses using both mouse models indicate that improvements in motor skills occurred with the C-6 treatment, especially at the high dosage of 25 mg/kg.

The NOX-1, -2 and -4 among NOX isoforms are known to be expressed in the midbrain [6,17,43,44,45]. Along with being expressed in neuronal cells, NOX-1 and -2 are also abundantly expressed in non-neuronal cells including microglia [16,33]. However, NOX-2 is the main isoform in the NOX family, expressed in astrocytes, and its activation may be linked to the expression of other isoforms, NOX-1 and -4 [17,21,46]. All these isoforms generate superoxide when activated, and elevated expressions have been detected in neurodegenerative conditions [20,21,47]. Although the novel C-6 was demonstrated to be an effective inhibitor for the three main NOX isoforms: NOX-1, -2, and -4, our results suggest that PFF injection elevated the expression of NOX-1 and -2 in the striatum and vMB, which was reversed by the C-6. Therefore, the improved neuroprotection or recovery to some extent may be mainly derived from the inhibition of NOX-1 and -2 activity. This is supported by the fact that NOX-1 and -2 activation is also seen with MPTP treatment in mice [48,49]. In our results, NOX-1 was detected as two distinct bands of approximately 59 kD and 63 kD, which pertain to two glycosylated sites on the full-length protein. NOX-2 was detected around 54 kD as expected in molecular weight, based on the description from the manufacturer. In addition, it has been shown that NOX-2^-/-^ mice show higher dopamine uptake capacity, increased dopaminergic neuronal numbers in the SNc, and higher dopamine level in the striatum, as well as superior motor functions in behavioral analysis, when they are exposed to the toxic doses of LPS and α-synuclein, compared with WT mice [20,48,50,51]. It should be also noted that microglial NOX-2, but not neuronal NOX-2 production, increases the sensitivity of rotenone and paraquat in dopaminergic neurons in rat brains in chemically induced PD pathology [43,52,53]. Thus, we consider NOX-2 as a promising target for halting PD pathology [48,53]. Since the antioxidant properties of the C-6 were successfully halting neurodegeneration by the inhibition of NOX-1 and -2 expression under the protein aggregate-mediated oxidative stress, the C-6 can be a potent neuroprotective/-recovery agent that can be further investigated as a potential therapeutic for PD treatment. Currently, we are encouraged to improve the specificity of the compound for reducing potential high-dose detrimental effects and enhancing the efficacy in vivo.

## 4. Materials and Methods

### 4.1. Animals

All animal protocols for this study were conducted in accordance with the United States Public Health Service Guide for the Care and Use of Laboratory Animals; all procedures were approved by the Institutional Animal Care and Use Committee (IACUC) at Delaware State University. The wild-type α-synuclein overexpressing transgenic (L-61) C57BL/6 background female mice (25–30 g) were received from the University of California, San Diego [24,31,37]. The female mice were bred with C57Bl/6 male mice that were obtained from Charles Rivers (Wilmington, MA, USA). The pups were genotyped, and only the male mice were selected at 3 months for behavioral analysis for 2 weeks of the compound gavaging (Figure 4A). Due to the X-linked transgenic expression, only the male L-61 animals showed parkinsonian symptoms [31]. For behavioral tests, a single animal per polyacrylic cage was housed with access to food and water ad libitum and maintained in standard housing conditions, i.e., room temperature 24 ± 1 °C and humidity 60–65% with 12:12, light–dark cycle. The male and female mice used for PFF treatment were received from Charles Rivers (Wilmington, MA, USA) at the age of 10 months. After further housing the mice for 2 additional months, they were given PFF treatment at 12 months.

### 4.2. Cell Culture and Treatment

The N27 parental rat dopaminergic cell line was obtained from EMD Millipore (Burlington, MA, USA) and used for the in vitro analysis. Cells at passage number 10 or lower were used to assess the efficacy of the C-6 treatment. The cells were treated with preformed fibrils of α-syn (PFF) from StressMarq Biosciences (SPR-324C, Victoria, BC, CAN). Prior to PFF treatment, the cells were treated with 5 µg/mL of wheat-germ agglutinin (WGA) (L-9640, Sigma-Aldrich) in media and incubated for 1 h. The media containing WGA was added with 0.1M *n*-Acetylglucosamine (PHR-1432, Sigma-Aldrich) into each well at 1 µg/mL PFF, which was incubated for 24 h to induce cellular stress. The cells were first seeded in a 96-well plate or 6-well plate, depending on the type of assays. Each well in a 96-well plate was plated with 4000 cells, and a minimum of 4 wells were designated for each treatment (*n* = 4/group). After the NOX inhibitor (C-6) was dissolved in DMSO, various concentrations of the compound were incubated in the PFF treated media for 24 h. All assays were analyzed at the end of 24 h of the compound treatment, as previously reported [28,30].

### 4.3. Cell Viability Assay

The MTT assay is a colorimetric assay that is based on the conversion of water-soluble MTT (3-(4,5-dimethylthiazol-2-yl)-2,5-diphenyltetrazolium bromide) into formazan crystal by mitochondrial dehydrogenase enzymes, soluble in 100% DMSO. The resulting solution is purple in color, and the color intensity is proportional to the viable cells for each well in a 96-well plate. The color intensity in MTT assays was measured using a spectrophotometer (SpectraMax M5^e^, Molecular devices, Dover, DE, USA), at 570 nm with a reference wavelength of 630 nm. The cells were incubated with the MTT reagent (ab211091, Abcam) at 37 °C for 3 h, followed by solubilizing the formazan crystals in solution.

### 4.4. Cytotoxicity Assay

The lactate dehydrogenase (LDH) is a cytosolic enzyme that is released into the media during cellular stress and increased levels of cytotoxicity. The cytotoxicity assay was measured using reagents from the Pierce LDH Cytotoxicity Assay Kit (88954, Thermo). For the assay, 50 µL of media from each well was put into wells of a new 96-well plate. The LDH reagents were prepared according to the protocol provided, and 50 µL of the LDH reagent was mixed into each well. The total 100 µL of media containing LDH and its reagent were incubated at 37 °C for 30 min, followed by the application of a stop solution. The mixture was analyzed using a spectrophotometer (SpectraMax M5^e^, Molecular devices) at 490 nm with a reference wavelength of 680 nm.

### 4.5. ROS Levels and Thioflavin-T assessment

The reactive oxygen species (ROS) is a released byproduct during oxidative stress. On the other hand, thioflavin-T is an assay used to understand the level of protein aggregation that can be detrimental to cell survival. In order to assess the level of oxidative stress, the cells were incubated in media containing 5 µg/mL of the ROS reagent CellROX deep red reagent (C10422, Thermo Fisher Scientific) and 25 µg/mL of thioflavin-T reagent (CAS:2390–54-7, Acros Organics) at 37 °C for 30 min. The cells were then washed 3 times with warm phosphate-buffered saline (PBS) without Ca^2+^ or Mg^2+^ (137 mM NaCl, 2.7 mM KCl, 4.3 mM Na_2_HPO_4_, and 1.47 mM KH_2_PO_4_), and several images from each well were collected using EVOS FL Cell Imaging Systems (Invitrogen, MA, USA). A minimum of 2 wells were dedicated to each group, and the experiments were repeated 3–4 times independently (total replicated: *n* = 8–10/group). The intensity of red fluorescence corresponding to ROS levels and green labels corresponding to protein aggregation was analyzed by blinded raters using ImageJ.

### 4.6. Stereotaxic α-Synuclein PFF Injection and Brain Isolation

All C57BL/6 wild-type mice (both male and female) were obtained from the Charles River when they are 9–10 months old (Wilmington, MA, USA). The mice were aged until 12 months and stereotaxically injected with 2.5 µL of mouse α-syn PFF (2 μg/μL in PBS, 5 µg each side, SPR-324, StressMarq) in the dorsal striatum bilaterally, as reported previously [29,30,38].

### 4.7. Gavaging Schedule

The transgenic mice and the PFF-injected mice were given oral treatment at two different doses: 5 mg/kg or 25 mg/kg. All the mice were weighed before gavaging and then routinely monitored to check for any drastic weight loss due to the compound administration. The transgenic mice were also given a pre-gavaging assessment to compare any improvement in their motor functions pertaining to the compound treatment. Once the pre-gavaging assessment was completed for male mice aged 4.5 to 5 months, the mice in the treatment groups received two weeks of oral gavaging at two different doses of 5 mg/kg and 25 mg/kg (*n* = 9–10/group). The PFF-injected mice were also monitored for 3.5 to 4 months and administered with C-6 at the previously stated low and high doses for 5–6 weeks. The extended gavaging time was applied to understand whether the ongoing neuronal loss caused by PFF toxicity can be deterred with the compound administration.

### 4.8. Tissue Isolation

All mice were anesthetized in an isoflurane chamber, followed by intracardiac perfusion containing ice-cold 0.9% saline coupled with 150 mM NaCl/70% Ethanol. After decapitation, the entire brains were carefully isolated. The right hemisphere was kept in 4% paraformaldehyde for 48 h for post-fixing, followed by 30% sucrose solution. Blocks were prepared in OCT, and the entire block was sectioned for immunohistochemical analysis. The left hemisphere was region-specifically isolated (e.g., striatum and ventral midbrain) for Western blot analysis to assess the levels of protein expression [38,54].

### 4.9. Dopamine Quantification in LC–MS Analysis

The measurement of dopamine level was carried out using liquid chromatography coupled with triple quadrupole mass spectrometry (TSQ Quantum ultra EMR, Thermo Fisher Scientific, San Jose, CA, USA) at the Korea Basic Science Institute, Seoul. The standard and samples were separated on the reverse-phase column (ROC C18, 3.0 mm × 150 mm, 5 um, RESTEK) with the mobile phases A (0.2% formic acid in water (*v*/*v*)) and B (MeOH:ACN = 3:7 (*v*/*v*)) [55].The quantification of dopamine was operated with an ESI–MS system in selected reaction monitoring (SRM) mode (*m*/*z* 322.00 > 137.15 (CE 9%)). Vehicle and treatment (*n* = 6) groups were analyzed with an unpaired Student’s *t*-test. Significance was considered if the difference was below 0.05.

### 4.10. Behavioral Assays

#### 4.10.1. Hindlimb Clasping

This is often characterized by bat-like posture when mice are picked up by the tail, indicating they are experiencing severe motor deficits. A mouse was placed on a horizontal plane and held at 2 cm from the tail tip, then slowly to the height of 10 cm from the surface. A video was recorded for 15 s for each trial. A total of 3 trials were taken over 3 consecutive days and analyzed for the time in hindlimb clasping by blind raters.

#### 4.10.2. Crossbeam Foot Slips

All mice were habituated for two weeks, followed by five trials where a cage was placed at the narrow end of a crossbeam, and each mouse was pushed to cross the beam, which becomes narrower in width at regular intervals of distance. As a mouse begins to cross the narrowing crossbeam and move toward the cage, it can experience foot slips. A mouse with poor motor coordination slips more frequently than a healthy mouse with optimal motor coordination. Videos were taken from both the right and left sides, and each foot slip was counted and averaged to see whether there was a difference between healthy mice and symptomatic mice with poor motor coordination.

#### 4.10.3. Rotarod

Rotarod is a behavioral test used to measure motor functionality. For three consecutive days, the mice were habituated on the rotating rod comprising a horizontal plane rotating at a speed of 4 rotations per minute (rpm), increasing by 2 rpm every minute for 10 min or until they fell off the rod at 20 rpm. The data were collected over three consecutive days where the animals were placed on the rod at 4 rpm, and the rod increased the speed by 1 rpm every 16 sec for 10 min or until the mouse fell off. A longer latency to fall indicates an improved motor functionality.

#### 4.10.4. Pole Test

The mice were placed on a vertical pole with their face up and habituated to face down the pole to climb down for at least 5 days. On the day of testing, the time taken for a mouse to turn and position for the descend descent was noted, and the latency for the climb-down was counted. A mouse falling off or slipping down the pole without turning was considered a failure. Videos of the pole test were taken and analyzed by a blind rater to record the total climb-down or turn time. Minimal 5 trials were performed for each animal, and the average of 5 trials was considered as the final score for data presentation.

#### 4.10.5. Nesting

A 5 × 5 cm cotton nesting patch was placed in a clean cage housing a single animal. On the following day, pictures were taken of the condition of the nesting material utilized and analyzed in a blind manner. The ability of an animal to tear up the cotton nesting material and build a nest dictates the activity and motor coordination levels of the animal. Successful making of a nest was given 5 points as the highest score, and a barely touched nesting patch was considered a low score of 1 point. The points were represented on the graph to show any improvement in nesting capabilities across the groups.

#### 4.10.6. Grooming

Individual pictures of each animal were taken over 3 days and analyzed for grooming capabilities. Additional data were collected over 6 weeks during the gavaging period for PFF-injected mice to monitor whether the mice in all groups performed grooming activities uniformly. The points were averaged at the end of the assessment after the mice were euthanized. The coat conditions of each animal were assessed to correlate motivation and activity levels for each animal. Dull and non-shiny coats detected showed severe motor deficits, whereas healthy animals spent more time grooming. The grooming pictures were also analyzed by blind raters and given a point out of 10, with 1 being a poor/dull coat and 10 being a shiny and healthy-looking coat.

### 4.11. Immunohistochemistry

The serial coronal sections of the right hemisphere were collected throughout the striatum and substantia nigra. Each section with a thickness of 14 µm was mounted as a set of 5 onto positively charged slides (Midwest Sci, Valley Park, MO, USA). The slides were kept at −80 °C until immunohistochemistry was performed. The sections were first thawed at room temperature for 30 min, followed by a rehydration step in 0.1M PB for 10 min. Afterward, the permeabilization step was performed using pre-chilled 100% acetone for 10 min. The excess acetone was dried, and permeabilization solution (40 µL Triton-X in 2% BSA) was added to the slides and kept aside for an additional 10 min. A blocking reagent containing 5% BSA in PB-T was given in the blocking step after drying the slides of excess permeabilization solution. The primary antibody was exposed under the coverslips to prevent the dehydration of the primary antibody. After the primary antibody (EMD Millipore-AB152) treatment at 4 °C overnight, the tissues were washed in PBS-T for 5 min three times. Then, slides were treated with a secondary antibody of Alexa-647-conjugated goat anti-rabbit (1:1000; Molecular Probes, Thermo Fisher Scientific) and incubated at room temperature for 2 h. The secondary antibody was washed in PBS-T for 3 × 5 min. Coverslips were mounted on slides using ProLong Diamond antifade mounting medium (Thermofisher, MA, USA). All images were captured using EVOS FL Cell Imaging Systems (Invitrogen), and every fifth section was analyzed for measuring the intensity of TH staining in the striatum and counting TH+ neurons in the SNc in a blinded manner. The same procedure was used for assessing the levels of phosphorylated Ser-129-α-synuclein (1:500, Cell signaling, cat# 23706S) and protein aggregation using thioflavin-T (20 µM, Acros). Pictures for specific regions such as the striatum and ventral midbrain (vMB) were taken using confocal microscopy and stitched together to obtain a single image for representation purposes. The intensities were measured using ImageJ by a blind rater.

### 4.12. Immunoblot Analyses

A total of three protein extractions were performed. In the first extraction, a gentle RIPA buffer containing the low concentration of detergent (25 mM Tris-HCL, 150 mM NaCl, 1% NP-40, 1% sodium deoxycholate and 0.1% SDS) was added with phosphatase inhibitors (100 µL for 10 mL) and 5 mM EDTA, as published by our group [28]. The mixture was sonicated and centrifuged at 14,000 rpm for 10 min. The supernatant was collected and used for performing mass spectrometry analysis and measuring the levels of α-syn proteins for different groups. The pellet was re-dissolved for the second extraction in the RIPA buffer containing 2% SDS, compared with 0.1% SDS, to further solubilize the insoluble phosphorylated α-syn proteins [29,41,56]. It was then centrifuged for the third time, and the resulting supernatant was used to load on precast polyacrylamide gel (SurePAGE Bis-Tris, 8–15%, 12 wells; GenScript) and transferred to PVDF membrane (Immobilon-P, EMD Millipore) using the Bio-Rad transfer apparatus. The membranes were further used for estimating the protein levels for each group. The primary antibodies pertaining to each protein analyzed were diluted at 1:1000 (p-Ser-129-α-synuclein, Cell signaling, cat# 23706S; NOX-1,2 rabbit antibody, Proteintech 17772–1-AP and 19013–1-AP) and incubated at 4 °C overnight. On the following day, the membranes were washed and treated with the secondary antibody (1:2000, Invitrogen-31462) at RT for 2 h, prior to developing the blots in the Immobilon Forte Western HRP substrate (WBLUF0500, EMD Millipore) for the detection under the ChemiDoc iBright CL1000 (Invitrogen 1000). Equal loading was estimated by stripping the membrane and re-probing against β-actin (1:4000; Invitrogen MA5-15739-HRP) and then incubated with secondary (1:8000, Invitrogen-31432) for 2 h. Molecular weights for each protein were checked using the PageRuler pre-stained protein ladder on the immunoblots (EMP Millipore-MPSTD4; Thermofisher-26616). ImageJ was used to analyze specific bands on each blot and measured against the loading control (β-actin) for normalizing each band intensity.

### 4.13. Statistical Analyses

In most data analyses, one-way ANOVA Dunnett’s test was applied to assess the effects of the C-6 treatment, compared with MPP+ or PFF-only-treated group (Figure 1, Figure 2 and Figure 3 and Figure 6, Figure 7, Figure 8, Figure 9, Figure 10, Figure 11 and Figure 12). In Figure 4, Student’s paired *t*-test was applied to compare before the treatment (pre-gavage) with after the treatment (post-gavage) for assessing the efficacy of the C-6 in behavioral tests, since the same animals were compared in effects. In Figure 5, Student’s unpaired t-test was performed to compare the level of dopamine from transgenic mice with vehicle or the treatment (*n* = 7–8/group). When *p*-values are below 0.05, the results are considered significant (*).

## Figures and Tables

**Figure 1 ijms-23-04262-f001:**
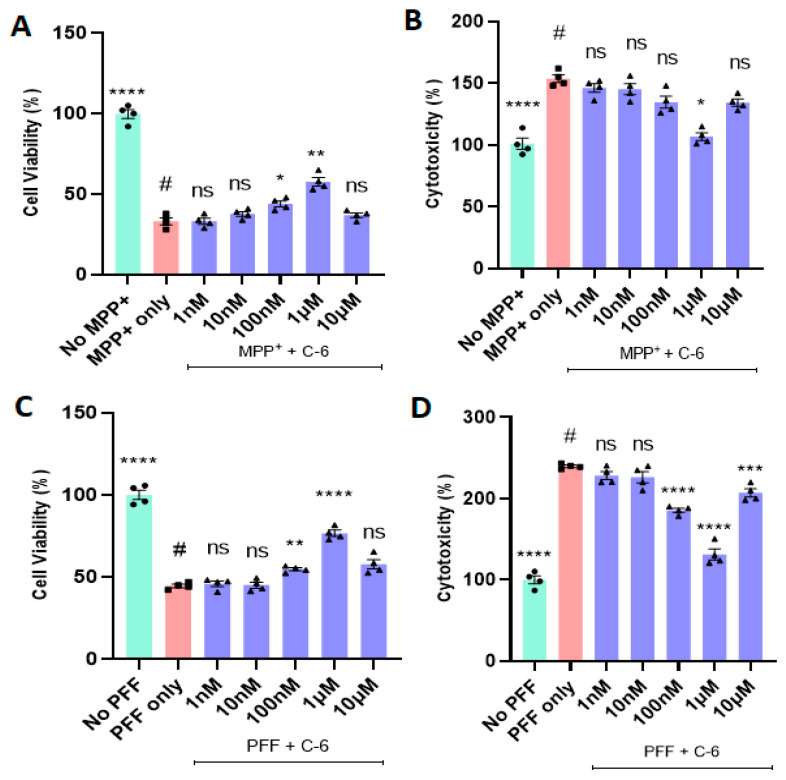
The treatment of a novel NOX inhibitor increases cell viability and decreases cytotoxicity derived from MPP+ or PFF in N27P dopaminergic cells. The optimal dose of the compound-6 was screened at various concentrations (1 nM–10 µM) in MTT (**A**,**C**) and LDH assay (**B**,**D**). The most effective concentration of the C-6 was consistently 1 µM against 640 µM MPP+ (**A**,**B**) and PFF (**C**,**D**) in both assays. (**A**,**B**) The C-6 increased cell viability and reduced the level of LDH (cytotoxicity) derived from MPP+ at the optimal dose of 1 µM. (**C**,**D**) The C-6 treatment increased the cell viability and decreased the cytotoxicity (LDH level) induced by PFF at the concentration between 100 nM and 1 µM. One-way ANOVA with Dunnett’s test was applied for statistical significance, compared with MPP+ only (#) or PFF only control (#) (*n* = 4/group independently). The no MPP+ (or no PFF) group is considered as positive control and converted to 100% for showing relativity. *: *p* < 0.05; **: *p* < 0.01; ***: *p* < 0.001, ****: *p* < 0.0001 and ns: not significant.

**Figure 2 ijms-23-04262-f002:**
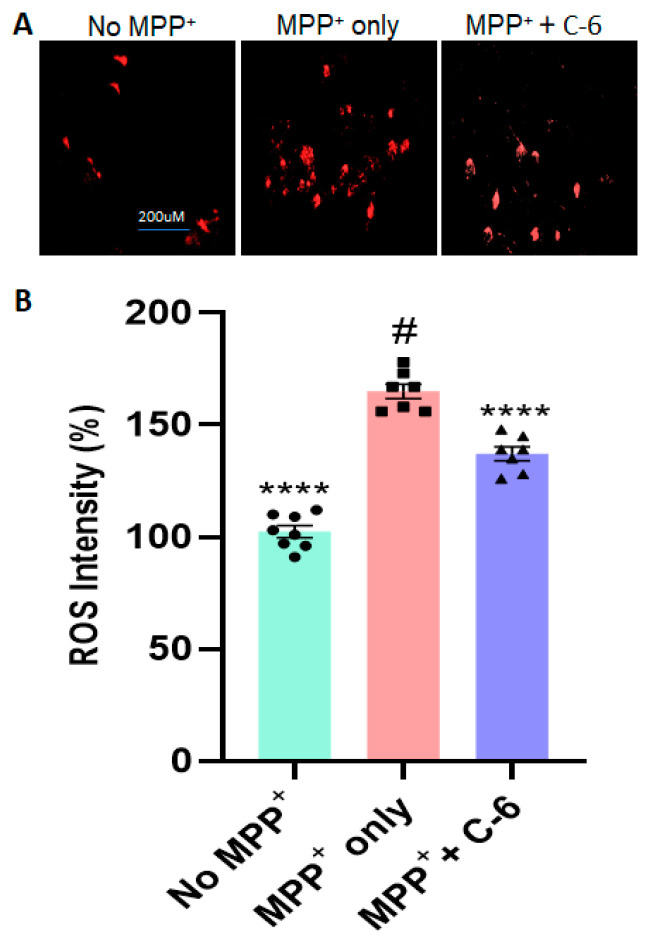
The compound-6 reduces the level of ROS generated by MPP+-induced toxicity in N27 cells. (**A**) The example images show that MPP+ exposure triggered a significant increase in ROS generation, which was reversed by the compound treatment, compared with no-MPP+ group. (**B**) The C-6 treatment (1 µM) markedly decreased the MPP+-induced ROS generation. One-way ANOVA with Dunnett’s test was adopted to compare with MPP+ only control (#) (*n* = 2–3 wells/experiment for 3 independent trials). ****: *p* < 0.0001.

**Figure 3 ijms-23-04262-f003:**
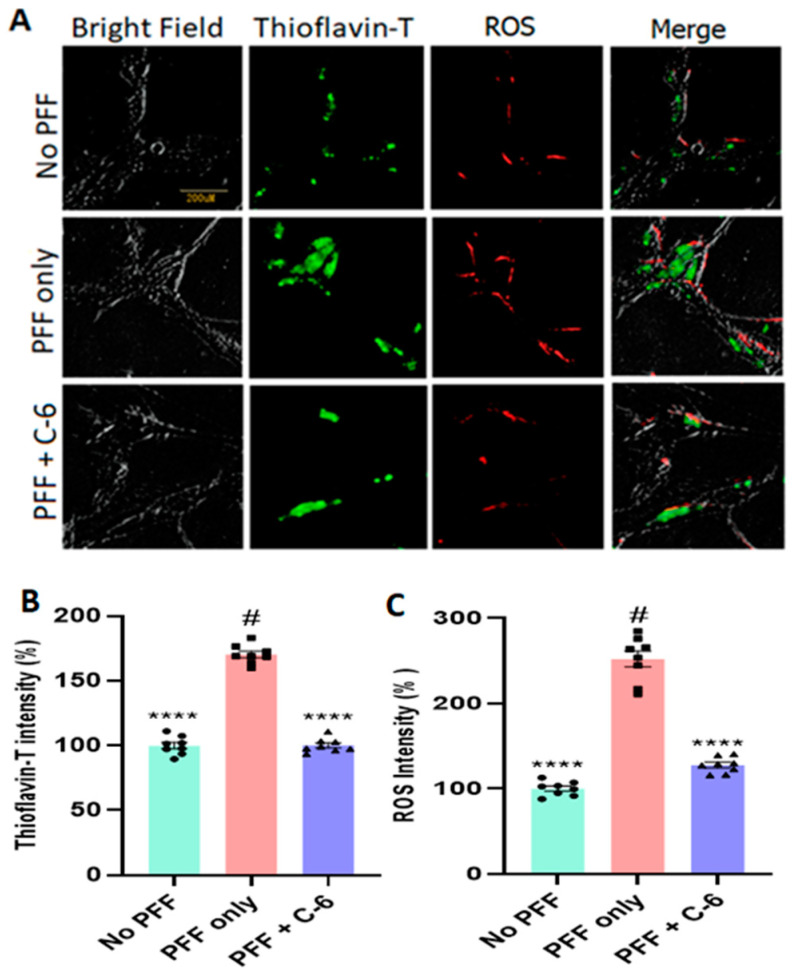
The C-6 effectively alleviates PFF-induced ROS generation and protein aggregation. (**A**) The example images support that PFF-induced ROS generation and protein aggregates were substantially reversed by the C-6 treatment, compared with no PFF. (**B**,**C**) The C-6 treatment significantly ameliorated both protein aggregation and ROS generation, induced by PFF exposure. The optimal concentration used was 1 µM, at which ROS generation and protein aggregation were significantly reduced with the C-6. One-way ANOVA with Dunnett’s test was applied to compare the effect of the treatment with PFF only (#) (*n* = 3 or 4 independently). ****: *p* < 0.0001.

**Figure 4 ijms-23-04262-f004:**
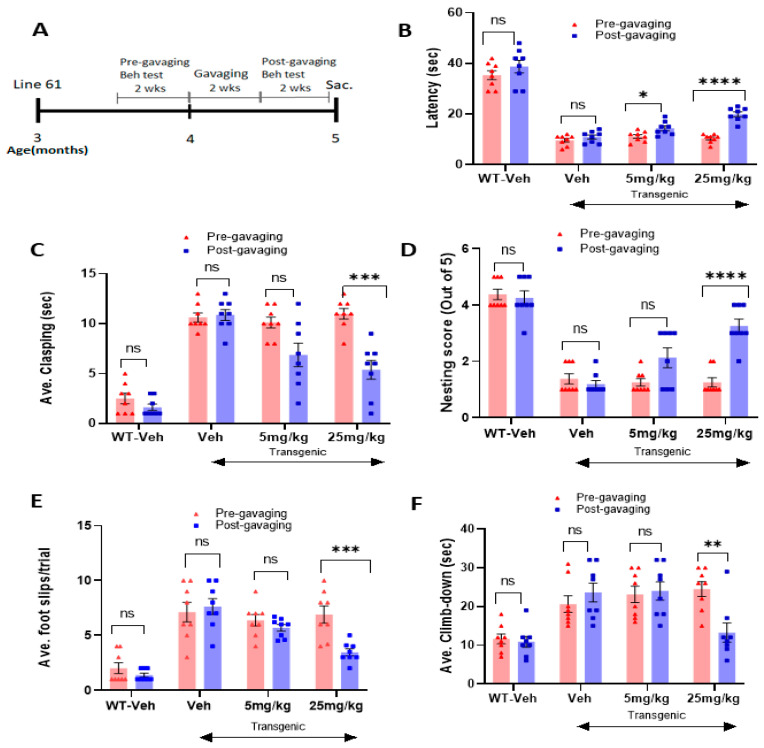
The oral treatment of the compound-6 for 2 weeks alleviates PD-like motor deficits in the L-61 transgenic mouse model. (**A**) the schematic diagram shows the gavaging and behavioral test schedule in timeline after transgenic mice turned 3 months old. The oral treatment improved the latency on rotarod (**B**), alleviated hindlimb clasping (**C**), enhanced nesting capabilities (**D**), reduced the slips on the crossbeam (**E**), and decreased the climb-down time on the pole test (**F**). Student’s paired *t*-test was performed to assess the effects of the compound treatment. Before treatment was labeled as pre-gavaging and after treatment as post-gavaging (*n* = 7–8/group). *: *p* < 0.05, **: *p* < 0.01, ***: *p* < 0.001 and ****: *p* < 0.0001. ns: not significant.

**Figure 5 ijms-23-04262-f005:**
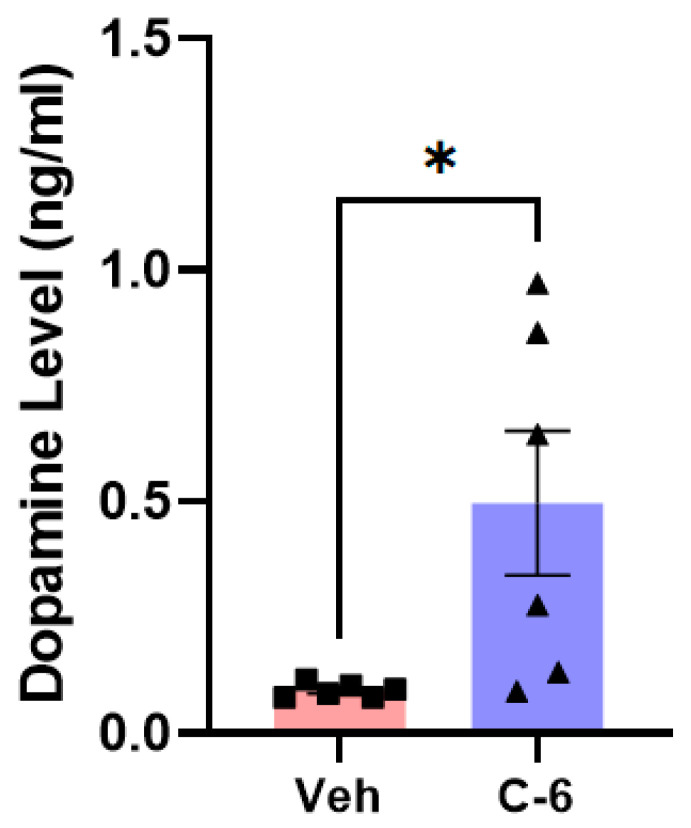
Two weeks of the compound-6 gavaging significantly increases the level of dopamine in the striatum of the L-61 mice. The treatment of 25 mg/kg substantially increased the dopamine released into the striatum. Unpaired Student’s *t*-test was applied to compare the treatment and vehicle groups (*n* = 6/group). *: *p* < 0.05.

**Figure 6 ijms-23-04262-f006:**
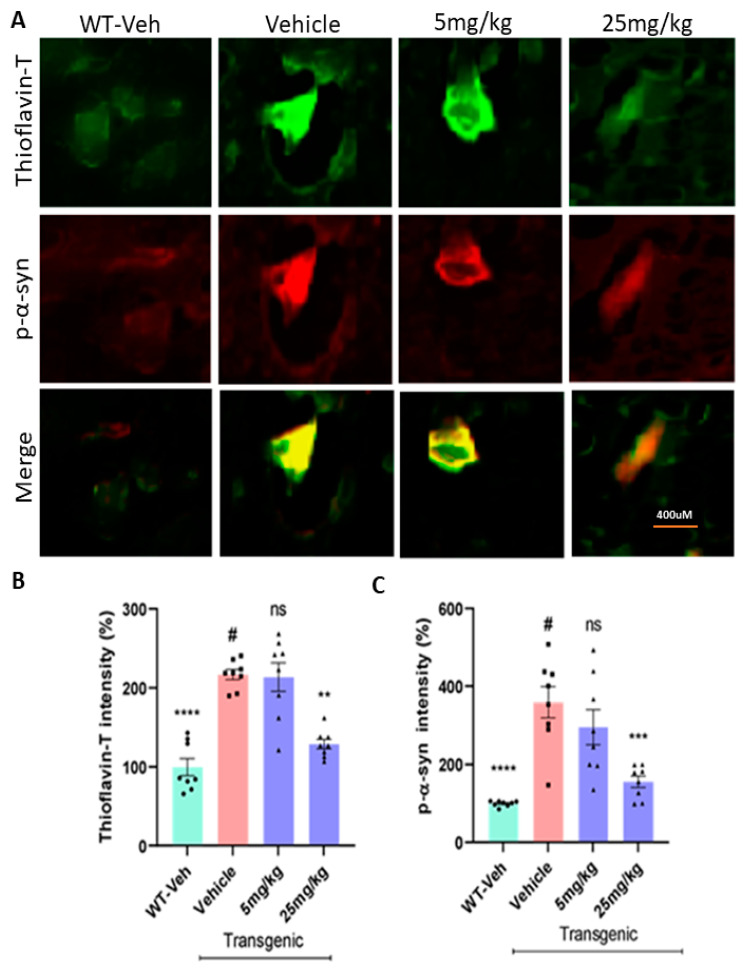
The C-6 treatment decreases the levels of protein aggregation and phosphorylated α-synuclein in the L-61 mice. (**A**) A few example images show the different levels of protein aggregates in Thioflavin-T stain and phosphorylated Ser-129-α-synuclein in the striatum. The oral treatment reduced the levels of protein aggregates in Thioflavin-T (**B**) and phosphorylated Ser-129-α-synuclein (**C**) in the striatum of transgenic mice. One-way ANOVA with Dunnett’s test to compare the intensities of Thioflavin-T stain and phospho-Ser129-α-synuclein label present across the groups with or without treatment (#) (*n* = 7–8/group). **: *p* < 0.01, ***: *p* < 0.001 and ****: *p* < 0.0001. ns: not significant.

**Figure 7 ijms-23-04262-f007:**
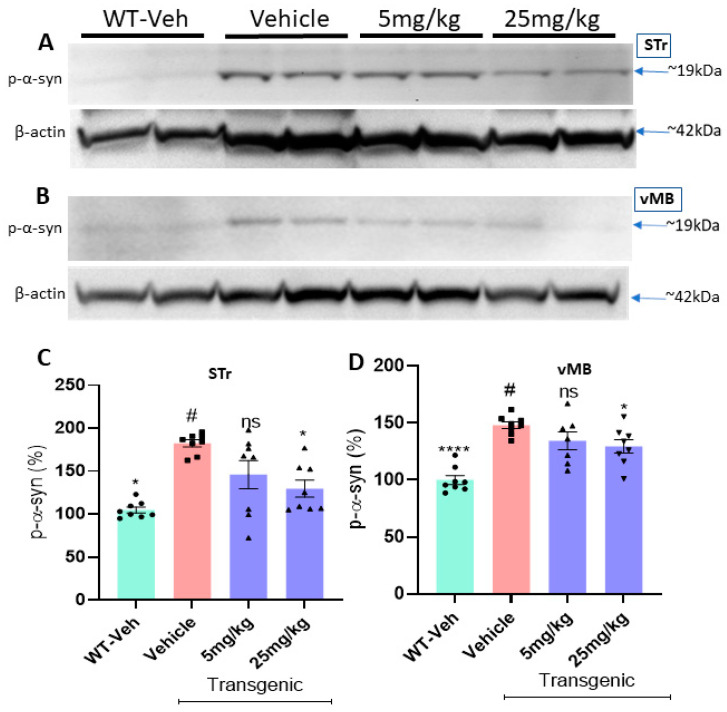
The compound treatment reduces the levels of phosphorylated Ser-129 α-synuclein in the striatum and the ventral midbrain of L-61 mice. (**A**,**B**) In Western blots, the oral treatment of the C-6 lowered the level of phosphorylated Ser-129 α-synuclein in the STr and vMB in a dose-dependent manner. The band intensity was also compared with vehicle-treated WT mice (WT-veh). (**C**,**D**) The oral C-6 treatment decreased the levels of phosphorylated Ser-129-α-synuclein in the striatum (**C**) and the ventral midbrain (vMB) (**D**). One-way ANOVA with Dunnett’s test was used to compare the band intensities of phospho-Ser129-α-synuclein among the treatments, vehicles (#), and vehicle-treated WT (*n* = 7–8/group). *: *p* < 0.05 and ****: *p* < 0.0001. ns: not significant.

**Figure 8 ijms-23-04262-f008:**
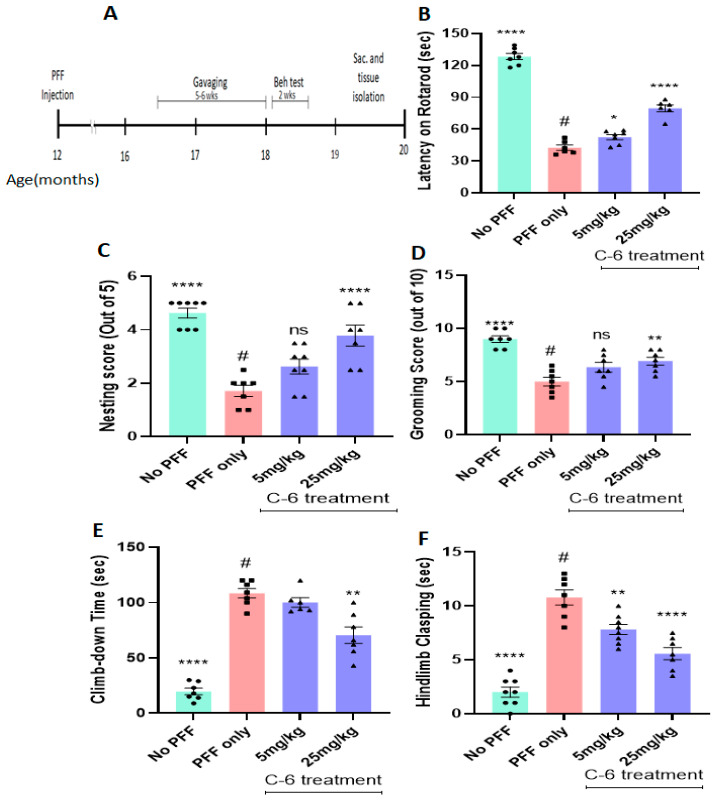
The compound-6 treatment alleviates motor dysfunctions in PFF-injected mice. (**A**) The schematic diagram shows the timeline of gavaging and behavioral test scheduled in PFF-injected mice. The oral treatment for 5–6 weeks improved the latency on rotarod (**B**), enhanced nesting capabilities (**C**), improved grooming (**D**), reduced the climb-down time on the pole test (**E**), and alleviated hindlimb clasping (**F**). One-way ANOVA with Dunnett’s test was applied to compare the treated groups with that of PFF-only mice (#) (*n* = 7–8/group). *: *p* < 0.05, **: *p* < 0.01 and ****: *p* < 0.0001. ns: not significant.

**Figure 9 ijms-23-04262-f009:**
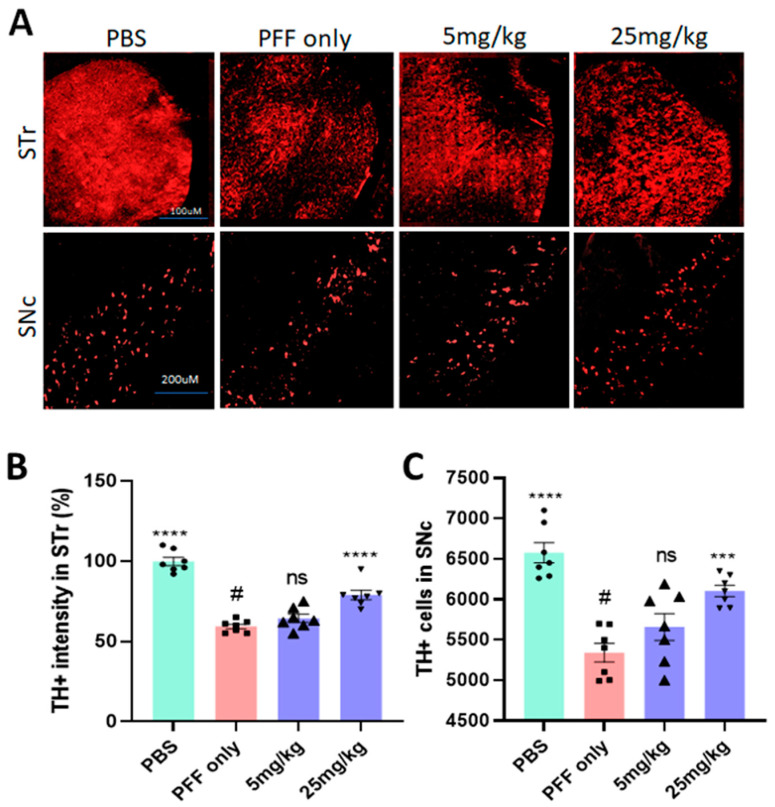
The C-6 treatment prevents or even recovers from dopaminergic neuronal damage in the SNc and increases TH intensity in the STr. (**A**) The examples of IHC show an increase in TH intensity in the STr and higher number of TH+ neurons in the SNc from mice treated with 25 mg/kg dose, compared with PFF only. (**B**) The C-6 treatment at 25 mg/kg enhanced the level of TH+ in the striatum, compared with PFF only (*n* = 7–8/group). (**C**) The number of TH+ neurons in the SNc is significantly higher with the C-6 treatment than PFF only (*n* = 7–8/group). One-way ANOVA with Dunnett’s test was used to compare the intensity of TH+ cells in the striatum and TH+ cell number in the SNc, compared with PBS-vehicle- or PFF-vehicle (#)-treated mice. ***: *p* < 0.001 and ****: *p* < 0.0001. ns: not significant.

**Figure 10 ijms-23-04262-f010:**
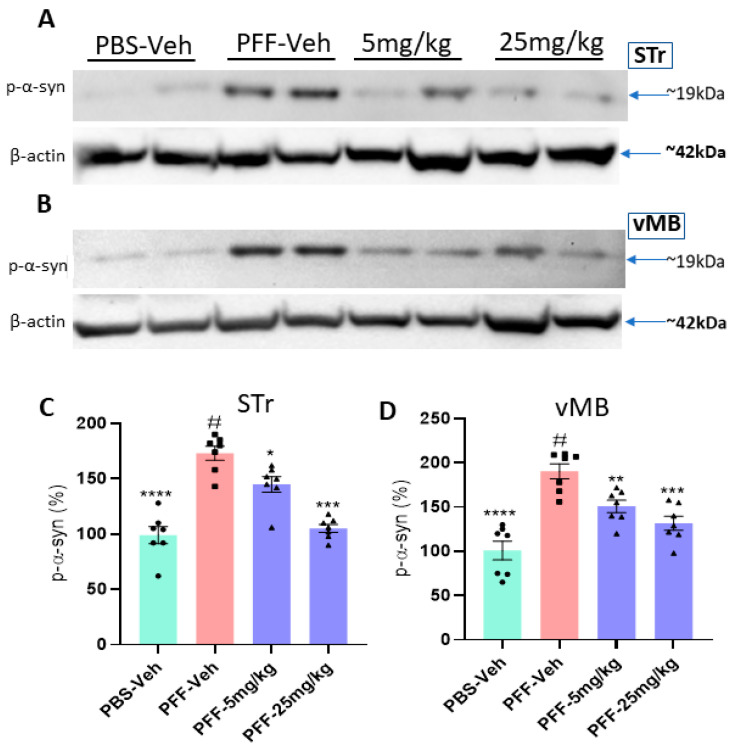
The C-6 treatment reduces the level of phosphorylated Ser-129-α-synuclein in the striatum and the ventral midbrain in PFF-injected mice. (**A**,**B**) As revealed by Western blots, PFF injection increased the level of phosphorylated Ser-129-α-synuclein, whereas the compound reduced the level in STr (**A**) and vMB (**B**). (**C**,**D**) The oral treatment of the compound-6 at both 5 mg/kg and 25 mg/kg significantly reduced the levels of phosphorylated Ser-129-α-synuclein in the striatum (**C**) as well as in the vMB (**D**) in a dose-dependent manner. One-way ANOVA with Dunnett’s test was applied to compare the expression level of phospho-Ser129-α-synuclein among the groups with or without treatment (#) (*n* = 7–8/group). *: *p* < 0.05, **: *p* < 0.01, ***: *p* < 0.001 and ****: *p* < 0.0001.

**Figure 11 ijms-23-04262-f011:**
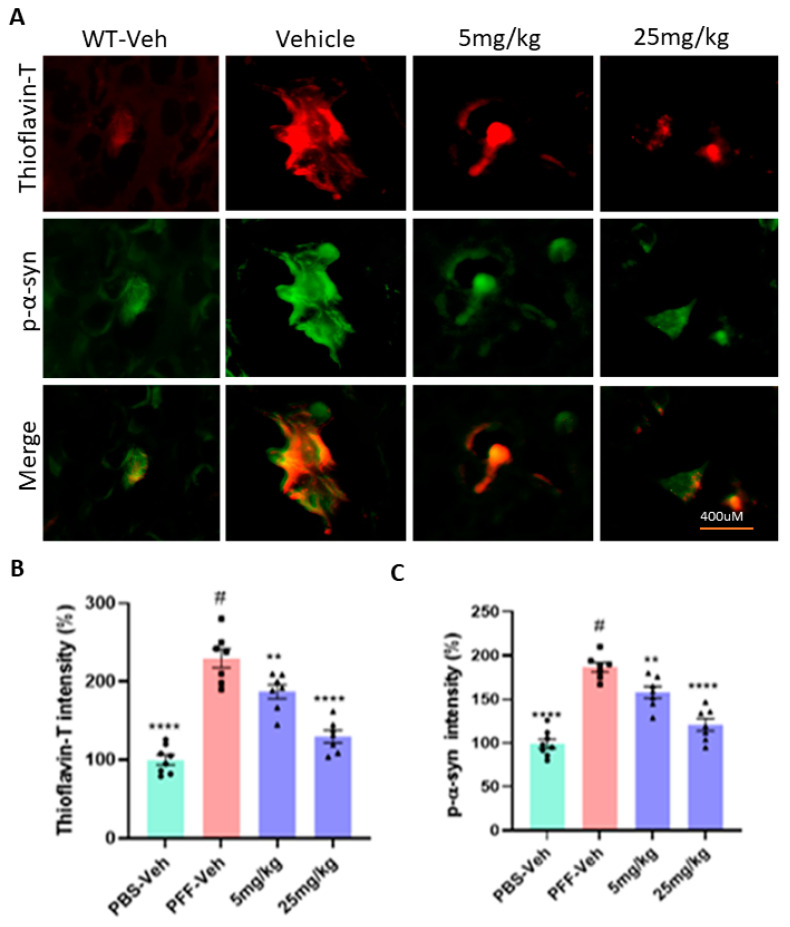
In the immunohistochemical analyses, the C-6 treatment decreases the levels of protein aggregates and phosphorylated Ser-129-α-synuclein in the striatum of PFF-injected mice. (**A**) Based on fluorescent co-labeling in immunohistochemistry, PFF injection enhanced the levels of protein aggregates in Thioflavin-T stain and phosphorylated Ser-129-α-synuclein, whereas the C-6 treatment substantially reduced both labeling intensities in the striatum. (**B**,**C**) In the analysis of Thioflavin-T label intensity, both 5 mg/kg and 25 mg/kg treatments significantly reduced the label intensity of protein aggregates (**B**) and phosphorylated Ser-129-α-synuclein (**C**) in the striatum. One-way ANOVA with Dunnett’s test was adopted to compare the intensity of Thioflavin-T stain and phospho-Ser129-α-synuclein label present across the groups with or without treatment (#) (*n* = 7–8/group). **: *p* < 0.01 and ****: *p* < 0.0001.

**Figure 12 ijms-23-04262-f012:**
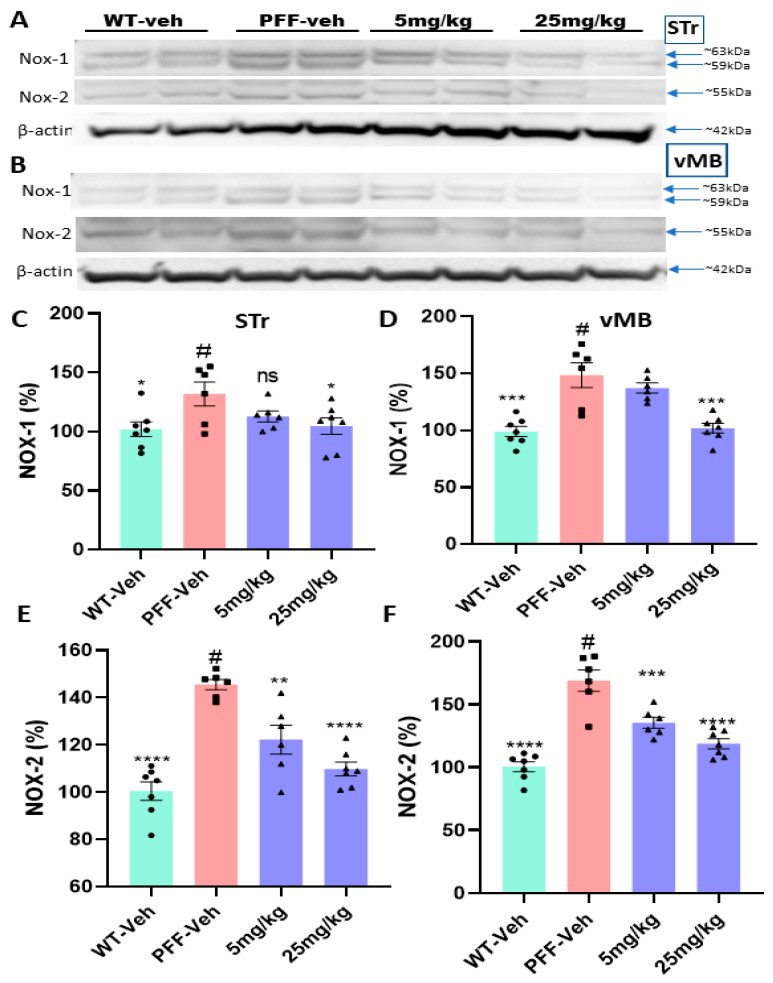
The C-6 treatment reduces the expression levels of NOX-1 and NOX-2 in the STr and vMB in Western blot analyses. (**A**,**B**) The examples of WB images show that both NOX-1 and -2 were upregulated by PFF injection, while the C-6 treatment reduced both levels in the STr (**A**) and vMB (**B**) in a dose-dependent manner. (**C**,**D**) The elevated NOX-1 level by PFF injection was reduced by the compound treatment in the STr (**C**) and vMB (**D**). (**E**,**F**) The PFF injection significantly increased the level of NOX-2, which was reversed by the C-6 treatment in the STr (**E**) and vMB (**F**). One-way ANOVA with Dunnett’s test was adopted to compare the expressions of NOX-1 or -2 across the groups with or without treatment (#), compared to WT-Veh (*n* = 7–8/group). *: *p* < 0.05, **: *p* < 0.01, ***: *p* < 0.001 and ****: *p* < 0.0001. ns: not significant.

**Table 1 ijms-23-04262-t001:** The compound-6 effectively inhibits NOX-1, -2 and -4, based on IC_50_ analysis.

Isoenzymes	NOX-1	NOX-2	NOX-4	NOX-5
C-6	0.42	0.35	0.34	6.87
(IC50: µM)				

## Data Availability

All data generated for the study are included in this published article and its Appendix A.

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
