# Peer review of "A Novel NOX Inhibitor Treatment Attenuates Parkinson’s Disease-Related Pathology in Mouse Models"

_ijms, 2022, doi:10.3390/ijms23084262_

Round 1
Reviewer 1 Report
This is a study, by Ghosh et al, evaluating the efficacy of a NOX inhibitor (C-6) in dopaminergic cells and PD mouse models.
The study is comprehensive and interesting.
I have the following comments:
Authors suggest that NOX inhibition with this compound can be an effective therapeutic application for PD pathology. Perhaps this is a excessive conclusion.
Sometimes C-6 is used and other compound-6
Authors say that C-6 enghances the cell viability and reduces the cytotoxicity from MPP+ or PFF in a dose dependent manner, up to 1 mM, but really there is no effect down to 100nM.
About behavior. What score scaling has been used?
About IHQ. Photos are not clear.
About WB. There are two bands in NOX1 but also in NOX2 (but authors cut it). Why?
In Figure 2. Authors say: “n=2-3 for 3 independent experiments”. That’s a very low n and 2 or 3 is very important. Authors must clarify this.
In my opinion, the sentence "The no MPP+ (or no PFF) is considered as positive control and converted to 100% for showing relativity" is incorrect. Reallt this is the reference group.
The picture is incorrect. It looks like that neurodegeneration produce the treatment with C-6
Author Response
Reviewer 1:
Comments and Suggestions for Authors
Open Review
This is a study, by Ghosh et al, evaluating the efficacy of a NOX inhibitor (C-6) in dopaminergic cells and PD mouse models.
The study is comprehensive and interesting.
I have the following comments:
- Authors suggest that NOX inhibition with this compound can be an effective therapeutic application for PD pathology. Perhaps this is a excessive conclusion.
As a reviewer suggested, we tone-downed our conclusion as below:
“this compound has a great potential to be used as an effective therapeutic to halt PD pathology”.
- Sometimes C-6 is used and other compound-6
We clearly see the inconsistency and changed the compound-6 to “C-6” consistently in the revised manuscript.
- Authors say that C-6 enhances the cell viability and reduces the cytotoxicity from MPP+ or PFF in a dose-dependent manner, up to 1 mM, but really there is no effect down to 100nM.
We agree with a reviewer that lower concentrations such as 1 and 10 nM did not show significant effects on both tests. Even though it was already mentioned that “The C-6 significantly mitigated the PFF-induced cytotoxicity in the range of concentrations (100 nM to 1 µM) consistently in both MTT and LDH assays”, we did not clarify that low concentrations (10 nM or lower) were not effective against MPP+. Thus, we added a phase highlighted in the Results section 2.1. “The reduced cell viability was rescued by the C-6 treatment in a dose-dependent manner, up to 1 µM, although 10 nM or lower were not significant (Figure 1A), and the same results were obtained in the cytotoxicity analysis (Figure 1B).”
- About behavior. What score scaling has been used?
As described in the method (4.10), the nesting scores were established as a 5-point nest rating scale, which was reflected in Figures 4D & 8C. In this rating system, 1 is the lowest score for untouched nesting material, and 5 is a completely dispatched 3-dimensionally built nest. Pictures were taken and blind raters were asked to give the scores as per the established nesting scoring. Grooming test was assessed in a similar way as nesting test, but its scale was up to 10 as described in 4.10. Figure 8D shows the scale of scores as well.
- About IHQ. Photos are not clear.
The displayed IHC images were obtained from stitched tile images using confocal microscopy. Since those images look patchy, we replace them with seamless images using EVOS fluorescent microscopy.
- About WB. There are two bands in NOX1 but also in NOX2 (but authors cut it). Why?
The antibody used for detection of NOX2 can pick up additional signals between 58kDa to 65kDa pertaining to glycosylation sites of NOX2. We only showed the lower band as it was the main band observed as per the manufacturer. The upper band was also 59kDa which was the same as the primary band that was also observed in NOX1. In order to avoid additional ambiguity since the lower band of NOX1 (59 kD) and glycosylated upper band (58 kD) are in similar sizes, we focused on analyzing the main non-glycosylated band for NOX2 to avoid the ambiguity between NOX1 and glycosylated band in the Western blot analysis.
- In Figure 2. Authors say: “n=2-3 for 3 independent experiments”. That’s a very low n and 2 or 3 is very important. Authors must clarify this.
We really apologize for the confusion. Each dot represented the average value of ROS intensities per 2 or 3 wells per experiment, which were performed in triplicate (n=3 independently). Therefore, we have obtained 7 or 8 values over 3 independent experiments for the combined analyses. We formatted this phrase in a different way: “(n=2-3/experiment for 3 independent trials)”.
- In my opinion, the sentence "The no MPP+ (or no PFF) is considered as positive control and converted to 100% for showing relativity" is incorrect. Really this is the reference group.
We agree with a reviewer on this point, however, we wanted to keep the No MPP+ or No PFF group as a positive control because we want to show the whole scale of each measurement including cytotoxicity, ROS intensity, Thioflavin-T level, etc. In addition, we would like to highlight the toxic effects induced by PFF that contains the pre-treatment of WGA and GluNAc, which didn’t produce any additional detrimental effect on the cells. This presentation also demonstrated the scale of damage induced by PFF treatment across the manuscript. We want to keep consistent pattern in the scale of the vehicle-treated group (No MPP+/No PFF) as 100%.
- The picture is incorrect. It looks like that neurodegeneration produce the treatment with C-6.
Although we could select a higher contrast image for showing C-6 effect in Figure 2, we intentionally selected an image displaying the intensity of ROS between ‘no MPP+’ and ‘MPP+ only’, since the level of recovery with C-6 was not the same level as positive control (no MPP+). We believe that the displayed image represents the level of ROS reduction by C-6 in scale (30%).

Reviewer 2 Report
Very interesting job.
Two small questions and comments.
1. Figure 1, replace the term cytotoxicity with LDH level. Or add to the figure caption how cytotoxicity was assessed.
2. There is not enough discussion of the reasons for the decrease in the effectiveness of C6 when moving from 1 to 10 μM.
Author Response
Reviewer 2:
Very interesting job.
Two small questions and comments.
1. Figure 1, replace the term cytotoxicity with LDH level. Or add to the figure caption how cytotoxicity was assessed.
Thank you for the suggestion. We have added “the level of LDH”, next to the cytotoxicity in Figure 1 caption.
“(A-B) The C-6 increased cell viability and reduced the level of LDH (cytotoxicity) derived from MPP+ at the optimal dose of 1µM. (C-D) The C-6 treatment increased the cell viability and decreased the cytotoxicity (LDH level) induced by PFF at the concentration between 100 nM and 1 µM.”
There is not enough discussion of the reasons for the decrease in the effectiveness of C6 when moving from 1 to 10 μM.
We appreciate the valid point addressed by a review.
In most cases of testing novel compounds or drugs, we can often find an optimal range of efficacy where the compound exhibits a biological response. In this case, we can clearly see that the compound shows a clear decrease in PFF induced toxicity at 1 µM. However, we found lower or no effects at 10 µM, which is considered an over-dose related adverse effects. Therefore, we are fully aware of using this compound (C-6) in low or moderate range of doses in future studies.